

# Clinical impact of albumin in advanced head and neck cancer patients with free flap reconstruction—a retrospective study

Ming-Hsien Tsai[1], Hui-Ching Chuang[1,2], Yu-Tsai Lin[1,2], Hui Lu[1], Wei-Chih Chen[1,2], Fu-Min Fang[2,3] and Chih-Yen Chien[1,2]

[1] Department of Otolaryngology, Kaohsiung Chang Gung Memorial Hospital and Chang Gung University College of Medicine, Kaohsiung, Taiwan
[2] Head and Neck Oncologic Group, Kaohsiung Chang Gung Memorial Hospital, Kaohsiung, Taiwan
[3] Department of Radiation Oncology, Kaohsiung Chang Gung Memorial Hospital and Chang Gung University College of Medicine, Kaohsiung, Taiwan

Corresponding author
Chih-Yen Chien,
cychien3965@adm.cgmh.org.tw

## ABSTRACT

**Background**. Poor nutritional status among patients with advanced stage head and neck squamous cell carcinoma (HNSCC) is common. Albumin is a common indicator of nutritional status and has been shown to be a predictor of oncological outcomes and perioperative morbidity. This study aims to determine the prognostic value of the serum albumin level among patients with advanced HNSCC undergoing surgery with simultaneous free flap reconstruction.

**Methods**. A total of 233 patients with advanced head and neck cancer undergoing tumor resection and immediate microvascular free flap reconstruction in a tertiary referral center were enrolled retrospectively between January 2009 and December 2011. Statistical analyses including Pearson's chi-squared test were used to determine whether there was a significant difference between each selected clinical factors and postoperative major wound infection. Multiple regression analysis was performed to reveal the relationship between postoperative major wound infection and clinical factors. Kaplan–Meier curves and multivariate Cox regression were applied to analyse survival outcome for overall survival (OS), disease-specific survival (DSS) and disease-free survival (DFS).

**Results**. Postoperative serum albumin level ($p < 0.001$) and tumor location were both significantly associated with postoperative major wound infection ($p = 0.018$) in univariate analysis. Multiple regression analysis showed a higher risk of postoperative major wound infection among patients with postoperative hypoalbuminemia than in their counterparts (odds ratio [OR] 9.811, 95% CI [2.288–42.065], $p = 0.002$). Patients with a tumor located over the hypopharynx experienced increased risk of postoperative major wound infection (OR 2.591, 95% CI [1.095–6.129], $p = 0.030$). With respect to oncological outcomes, preoperative serum albumin level is a significant independent prognostic factor for overall survival (OS), disease-specific survival (DSS) and disease-free survival (DFS).

**Conclusions**. Postoperative hypoalbuminemia is a useful indicator for the development of postoperative complications. In addition, preoperative hypoalbuminemia is a negative prognostic factor for patients who have undergone tumor excision and free flap reconstruction for the advanced stage of HNSCC.

# INTRODUCTION

The nutritional status of patients with head and neck squamous cell carcinoma (HNSCC) can affect the disease course and prognosis (*Langius et al., 2013*). Patients with HNSCC are vulnerable to malnutrition at the time of diagnosis and during treatment. It may be caused by the tumor itself or by treatment-related symptoms such as dysphagia, odynophagia, anorexia, mechanical obstruction and fatigue (*Ravasco et al., 2003*). Generally, poor nutrition status affects treatment compliance, quality of life and survival outcomes in patients with cancer (*Ravasco et al., 2004*). Serum albumin concentrations may be related to weight loss among patients with cancer due to changes in body mass and systemic inflammatory reactions (*McMillan et al., 2001*). Serum albumin has also been investigated as a predictor of cancer survival and perioperative outcomes in a variety of cancers. Only a few studies have focused on clinical impact on the level of serum albumin in head and neck cancer patients (*Danan et al., 2016*).

Surgical intervention with adequate free margins is important for disease control in patients with head and neck cancer; however, a large defect can lead to great morbidity, dysfunction and disfigurement (*Chen et al., 2012*). Microvascular free flap transfer for head and neck reconstructive surgery has become increasingly popular during the past 20 years (*Dassonville et al., 2008*). However, when microsurgical complications occur, they not only delay postoperative adjuvant therapy but also result in subsequent treatment failure (*Jones et al., 2007*). Currently, data are still limited regarding the role of hypoalbuminemia among patients with HNSCC who have undergone surgery for tumor removal with free flap reconstruction. The aim of this study is to determine the prognostic value of serum albumin level among patients with advanced HNSCC who have undergone surgery for HNSCC and free flap reconstruction.

# MATERIALS AND METHODS

## Study population

This study retrospectively enrolled patients who were diagnosed with HNSCC and underwent surgical resection of tumor with microvascular free flap reconstruction simultaneously between January 2009 and December 2011 in Kaohsiung Chang Gung Memorial hospital, Taiwan. The inclusion criteria were age >20 years and advanced stage (stage III/ IV) of HNSCC. The patients underwent regular examinations of body weight and circulatory laboratory data before surgery. In total, 233 patients (220 men and 13 women) with a median age of 54 years (30–84 years) were included for analyses. Tumors were re-staged according to the eighth edition of the American Joint Committee on Cancer Staging (AJCC). Treatment was primarily based on the American National Comprehensive Cancer Network (NCCN) guidelines. Concurrent chemoradiation (CCRT) or radiation was conducted for these cases. The chemotherapy agent was cisplatin-based. The radiation

technique for these patients was intensity-modulated radiation therapy (IMRT). The primary radiation dose for HNSCC was between 6,800 and 7,000 cGy for gross disease. The mean adjuvant radiation dose was 6,000 cGy.

## Variables and outcomes

Patients were retrospectively enrolled according to the following clinical characteristics: gender, age, primary tumor site, cancer stage and follow-up information about recurrence and survival after treatment. Body mass index (BMI, kg/m$^2$) and circulatory laboratory data including serum albumin were regularly measured within one week before the surgery. The definition of postoperative serum albumin is the serum albumin level collected the morning following the surgery. Hypoalbuminemia is defined as serum albumin level <3.5 g/dL. Major postoperative wound infection is defined as a postoperative recipient-site wound condition that necessitated wound debridement in the operation room.

## Statistical analysis

Statistical analyses were performed using IBM SPSS Version 20.0 software (SPSS/IBM, Inc., Chicago, IL, USA). The end points of this study included postoperative wound infection, five-year disease-free survival (DFS), five-year disease-specific survival (DSS) and five-year overall survival (OS). Pearson's chi-squared test was used to determine whether there was a significant difference between each selected clinical factors and postoperative major wound infection. Multiple regression analysis was performed to reveal the relationship between postoperative major wound infection and clinical factors. The Kaplan–Meier method was utilized to estimate survival, and the log-rank test was used to examine statistical significance of factors. Cox proportional hazards models were used to identify significant variables associated with survival. The estimated hazard ratios (HRs) and 95% confidence intervals (CIs) were calculated. A two-tailed $p$ value < 0.05 was considered significant. This study was approved by the Medical Ethics and Human Clinical Trial Committees at Chang Gung Memorial Hospital (Ethical Application Reference number: 201701238B0).

## RESULTS

A total of 233 patients were enrolled in this study. The clinical characteristics of the study patients are summarized in Table 1. The median age of patients was 54 years (range: 30–84). The population included 220 (94.4%) male patients and 13 (5.6%) female patients. The average BMI in this population was 23.03 kg/m$^2$ (range: 14.7–42.6). The average preoperative serum albumin level was 4.2 g/dL (range: 2.5–5.8). The average postoperative serum albumin level was 3.1 g/dL (range: 1.9–4.3). In this cohort, 53 patients (22.7%) had type II diabetes mellitus and 24 patients (10.3%) had chronic renal disease. The sole histopathologic cancer type in this population was squamous cell carcinoma. The most common tumor subsite was the oral cavity ($n = 156$, 66.9%), followed by the oropharynx ($n = 31$, 13.3%), hypopharynx ($n = 27$, 11.6%) and larynx ($n = 19$, 8.2%), respectively. All oropharyngeal cancers in this cohort were p16 negative tumors. There were 59 patients who underwent salvage surgery for recurrent or persistent HNSCC, 49 of whom had had prior radiotherapy. The flap was most often harvested from the anterolateral thigh (ALT)

**Table 1** Clinicopathological characteristics of 233 patients with advanced head and neck cancer resection with free flap transfer.

| Characteristics | | Value | % |
|---|---|---|---|
| Median Age (year) [range] | | 54 [30, 84] | |
| Sex | male | 220 | 94.4 |
| | female | 13 | 5.6 |
| Average BMI (Kg/m$^2$) [range] | | 23.0 [14.7, 42.6] | |
| Average preoperative albumin (g/dL) [range] | | 4.2 [2.5, 5.8] | |
| Average postoperative albumin (g/dL) [range] | | 3.1 [1.9, 4.3] | |
| Diabetes mellitus | | 53 | 22.7 |
| Chronic renal disease | | 24 | 10.3 |
| Postoperative major wound infection | | 60 | 25.8 |
| Cancer type | SCC | 233 | 100.0 |
| Cancer stage | III | 45 | 19.3 |
| | IV | 188 | 80.7 |
| Salvage surgery | No | 174 | 74.7 |
| | Yes | 59 | 25.3 |
| Preoperative radiotherapy | No | 184 | 79.0 |
| | Yes | 49 | 21.0 |
| Cancer site | Oropharynx | 31 | 13.3 |
| | Larynx | 19 | 8.2 |
| | Hypopharynx | 27 | 11.6 |
| | Oral Cavity | 156 | 66.9 |
| Flap design | ALT | 229 | 98.3 |
| | Radial forearm | 1 | 0.4 |
| | AMT | 3 | 1.3 |
| Median follow up time (months) [range] | | 53.9 [0.8, 88.7] | |
| Recurrence | Yes | 75 | 32.2 |
| | No | 158 | 67.8 |
| Recurrent site | Local | 26 | 34.7 |
| | Regional | 30 | 40.0 |
| | Distant | 19 | 25.3 |
| Last status | NED | 154 | 66.1 |
| | DOD | 51 | 21.9 |
| | DWOD | 28 | 12 |

**Notes.**

BMI, body mass index; SCC, squamous cell carcinoma; ALT, anterolateral thigh; AMT, anteromedial thigh; NED, no evidence of disease; DOD, died of disease; DWOD, died without disease.

($n = 229$, 98.3%), followed by the anteromedial thigh (AMT) ($n = 3$, 1.3%) and radial forearm ($n = 1$, 0.4%). The incidence of postoperative major wound infection was 60 (25.8%). There was no postoperative donor site wound infection or complication in the population. Patients were followed up for a median of 53.9 months in this cohort. Tumor recurrence occurred in 75 (32.2%) patients, including local, regional and distant recurrence of 26 (34.7%), 30 (40.0%) and 19 (25.3%) patients, respectively. At the time of the last

**Table 2  Univariate analysis of risk factors impacting postoperative major wound infection of all patients.**

| Variable | | Number | Major wound infection | p |
|---|---|---|---|---|
| Stage | III | 45 | 9 | 0.326 |
| | IV | 188 | 51 | |
| Salvage surgery | No | 174 | 42 | 0.334 |
| | Yes | 59 | 18 | |
| Primary cancer site | Not hypopharynx | 207 | 48 | 0.018 |
| | Hypopharynx | 26 | 12 | |
| Preoperative serum albumin level (g/dL) | <3.5 | 16 | 6 | 0.253 |
| | ≧3.5 | 217 | 54 | |
| Postoperative serum albumin level (g/dL) | <3.5 | 187 | 58 | <0.001 |
| | ≧3.5 | 46 | 2 | |
| Age | <60 | 160 | 39 | 0.477 |
| | ≧60 | 73 | 21 | |
| BMI (Kg/m$^2$) | <22 | 96 | 24 | 0.826 |
| | ≧22 | 137 | 36 | |
| Diabetes mellitus | No | 180 | 47 | 0.817 |
| | Yes | 53 | 13 | |

follow-up, 154 (66.1%) patients remained disease-free, 51 (21.9%) patients had died of the disease and 28 (12%) patients had died of other diseases. The five-year OS rate in this cohort is 65.5%.

We were interested in the effects of the clinical factors, especially the albumin level affecting the postoperative major wound infection. The chi-square test for associations between each clinical factor and postoperative major wound infection were applied univariately and the *p*-values are listed in Table 2.

Several factors influencing postoperative wound infection were selected for univariate analysis (Table 2). Postoperative lower serum albumin level ($p < 0.001$) and primary cancer site over hypopharynx ($p = 0.018$) were both significantly associated with higher probability of postoperative major wound infection. To avoid the collinearity of predictors in a regression model, chi-square test of association between postoperative lower serum albumin level and primary cancer site over hypopharynx was performed. This analysis showed that there was no statistically significant association between the two predicting variables (chi-square test statistic $= 0.468$, $p = 0.494$). A multiple regression analysis then was applied to analyze the relationship between postoperative major wound infection and the two uncorrelated and significant factors revealed in prior univariate analyses. In this model, postoperative serum albumin level was a significant independent predictor of major wound infection (OR 9.811, 95% CI [2.288–42.065], $p = 0.002$). In addition, tumor location over the hypopharynx significantly increased the probability of postoperative major wound infection as compared to other cancer sites (OR 2.591, 95% CI [1.095–6.129], $p = 0.030$) (Table 3).

**Table 3** Regression analysis of factors impacting postoperative major wound infection of all patients.

| Factor | | Odds ratio | 95% Confident Interval | | p value |
|---|---|---|---|---|---|
| Postoperative albumin (g/dL) | ≧3.5 | 1 | | | .002 |
| | <3.5 | 9.811 | 2.288 | 42.065 | |
| Primary cancer site | Not Hypopharynx | 1 | | | .030 |
| | Hypopharynx | 2.591 | 1.095 | 6.129 | |

**Table 4** Univariate analysis of risk factors impacting survival of all patients.

| Variable | | Number | Event | OS (%) | p | Event | DSS (%) | p | Event | DFS (%) | p |
|---|---|---|---|---|---|---|---|---|---|---|---|
| Cancer stage | III | 45 | 11 | 75.8 | 0.053 | 6 | 85.7 | 0.064 | 10 | 76.7 | 0.056 |
| | IV | 188 | 69 | 61.6 | | 45 | 71.6 | | 65 | 62.7 | |
| Salvage surgery | No | 174 | 49 | 71.3 | <0.001 | 29 | 81 | <0.001 | 48 | 70.7 | 0.004 |
| | Yes | 59 | 31 | 44.8 | | 22 | 55.6 | | 27 | 50.2 | |
| Primary cancer site | Not Hypopharynx | 206 | 70 | 65.3 | 0.881 | 43 | 75.8 | 0.648 | 65 | 66.1 | 0.908 |
| | Hypopharynx | 27 | 10 | 56.6 | | 8 | 66.2 | | 10 | 61.8 | |
| Preoperative albumin level (g/dL) | <3.5 | 16 | 10 | 34.4 | 0.011 | 7 | 49.2 | 0.02 | 9 | 39.7 | 0.036 |
| | ≧3.5 | 217 | 70 | 66.9 | | 44 | 76.4 | | 66 | 67.4 | |
| Postoperative albumin level (g/dL) | <3.5 | 187 | 65 | 64.8 | 0.807 | 41 | 74.8 | 0.987 | 56 | 66.7 | 0.781 |
| | ≧3.5 | 46 | 15 | 62.6 | | 10 | 73.3 | | 16 | 61.8 | |
| BMI (Kg/m²) | <22 | 96 | 35 | 61.8 | 0.532 | 24 | 70.8 | 0.353 | 32 | 64 | 0.824 |
| | ≧22 | 137 | 45 | 66.3 | | 27 | 77.2 | | 43 | 66 | |
| Diabetes mellitus | No | 180 | 59 | 66.8 | 0.451 | 39 | 74.9 | 0.897 | 60 | 64.6 | 0.523 |
| | Yes | 53 | 21 | 56.7 | | 12 | 73.5 | | 15 | 68 | |

**Notes.**

OS, overall survival; DSS, disease specific survival; DFS, disease free survival.

In this study we also evaluated prognostic factors such as cancer stage, whether or not the patient had undergone salvage surgery, cancer site, pre- or post-operative hypoalbuminemia, body mass index (BMI) and diabetes mellitus that could have an impact on survival. Five-year survival rates of DSS, DFS and OS were calculated for all factors and are listed in Table 4. These results show that salvage surgery type (p value: OS: <0.001; DSS: <0.001; DFS: 0.004) and preoperative hypoalbuminemia (p value: OS: 0.011; DSS: 0.02; DFS: 0.036) are significantly associated with lower rates of survival of OS, DSS and DFS. Kaplan–Meier survival curves were drawn for two preoperative albumin levels (Fig. 1). The effects of hypoalbuminemia on survival were evaluated by fitting all the variables whose p-value < 0.25 in the Kaplan–Meier survival analyses to the Cox Regression model. The final analyses including the hypoalbuminemia variable for different survival outcomes were conducted, and the results are listed in Table 5. Only the preoperative hypoalbuminemia prognostic factor remained in the Cox Regression modeling. The results demonstrate that preoperative hypoalbuminemia was the independent prognostic factor of OS (hazard ratio [HR] 2.005, 95% CI [1.021–3.936], $p = 0.043$), DSS (HR 2.503–95% CI [1.126–5.563], $p = 0.024$) and DFS (HR 2.101, 95% CI [1.046–4.211], $p = 0.037$).

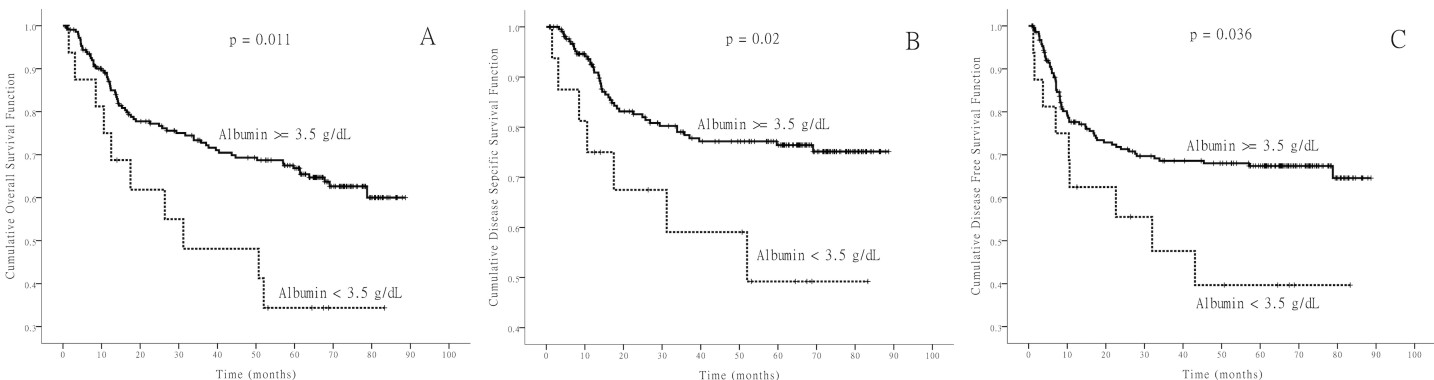

**Figure 1    Kaplan-Meier survival curves.** Kaplan-Meier survival curves of overall survival (A), disease-specific survival (B), and disease-free survival (C) according to the serum albumin level before operation.

**Table 5    Cox Regression Analysis (proportional hazard analysis) of prognostic factors to survival of all patients.**

| Factor | | Hazard ratio | 95% Confident Interval | | p value |
|---|---|---|---|---|---|
| *Proportional hazard analysis of overall survival (OS)* | | | | | |
| Preoperative albumin level (g/dL) | ≧3.5 | 1 | | | .014 |
| | <3.5 | 2.307 | 1.188 | 4.480 | |
| *Proportional hazard analysis of disease specific survival (DSS)* | | | | | |
| Preoperative albumin level (g/dL) | ≧3.5 | 1 | | | .024 |
| | <3.5 | 2.503 | 1.126 | 5.563 | |
| *Proportional hazard analysis of disease free survival (DFS)* | | | | | |
| Preoperative albumin level (g/dL) | ≧3.5 | 1 | | | .037 |
| | <3.5 | 2.101 | 1.046 | 4.211 | |

## DISCUSSION

Nowadays, microsurgical free flap transfer is the mainstay of reconstruction for large defects in the head and neck regions after tumor excision (*Ariyan, Ross & Sasaki, 1997*; *Lutz & Wei, 2005*). Successful head and neck reconstruction in cancer patients is crucial because delayed wound healing may adversely affect the oncologic outcomes of these patients and increase the length of hospital stay and cost. Several studies have investigated the importance of serum albumin level in postoperative wound infection among patients who have undergone head and neck cancer resection and simultaneous free tissue transfer. *Lo et al. (2017)* revealed that low albumin level was associated with a greater tendency to develop postoperative complications in 61 patients who underwent head and neck cancer surgery and simultaneous free tissue transfer. *Danan et al. (2016)* identified a significant correlation between preoperative hypoalbuminemia and increased probability of wound infection among patients with head and neck cancer. Another report also found that postoperative low albumin level is associated with postoperative complications after free flap transfer (*Hoppe, Abernathie & Datiashvili, 2012*). In agreement with these

studies, postoperative hypoalbuminemia carried a higher risk of postoperative major wound infection in our series. Basically, a decrease in serum albumin is associated with a decrease in important serum proteins of the immune system, which are essential for infection and tumor control. Although preoperative serum albumin level didn't show significant association with major wound infection by multivariate analysis in this cohort, the postoperative albumin level remained the significant factor affecting wound infection. Routine measurement of postoperative albumin levels and adequate supply of postoperative nutrition by tube feeding are advised to avoid potential wound complication and to shorten the hospital stay.

Primary cancer location is another variable affecting the postoperative wound infection, especially among patients of hypopharyngeal cancer undergoing total laryngopharyngectomy. The tubed free flap harvested from the anterior lateral thigh showed higher incidence of pharyngocutaneous fistula (PCF). This is likely a result of inadequate nutrition status during treatment, the complexity of the surgical procedure and early wound breakdown from the free flap. This type of fistula may further jeopardize the pedicle of the free flap, causing the loss of the free flap and even massive bleeding from the pedicle. The PCF may also intensify the psychological stress of patients, lead to serious wound complications, prolong the hospital stay, increase treatment costs and even hamper the patient's chance of survival.

The relationship between hypoalbuminemia and prognosis of treatment outcome among patients with cancer was noted in some studies. One systemic review in 2010 reported that higher serum albumin levels in patients would lead to higher rates of survival of gastrointestinal cancer (*Gupta & Lis, 2010*). There are also limited studies reporting the prognostic significance of pretreatment hypoalbuminemia in head and neck cancer patients. Advanced cancer stage is the strongest predictor of OS in patients with HNSCC. In patients with stage IV disease, advanced age and low serum albumin were associated with poorer OS (*Medow, Weed & Schuller, 2002*). A recent study revealed that preoperative hypoalbuminemia would lead to a higher probability of postoperative wound infection and poorer OS (*Danan et al., 2016*). Another study showed that pretreatment albumin level was one of the independent predictors of DFS, cancer-specific survival and OS. Patients with HNSCC showing hypoalbuminemia prior to treatment would experience approximately six-fold increases in the risks of tumor progression and cancer-specific and overall mortality (*Lim et al., 2017*). Our study also revealed similar results, showing preoperative hypoalbuminemia to be associated with poorer oncologic outcomes. These findings will prompt the clinicians to be cautious about the clinical impact of hypoalbuminemia in the surgical treatment of HNSCC.

## CONCLUSIONS

Postoperative hypoalbuminemia is useful as an indicator for the development of postoperative complications. In addition, preoperative hypoalbuminemia is a negative prognostic factor among patients who undergo tumor excision and free flap reconstruction for advanced stage of HNSCC.

### Funding

This work was supported by grants nos. CORPG8F1481, CMRPG8A0961, CMRPG8D1421-3, CMRPG8C0071, CMRPG8B0971-3, CMRPG8C0071, and CMRP100-0689B, CMRPG8E1472 from Kaohsiung Chang Gung Memorial Hospital. The funders had no role in study design, data collection and analysis, decision to publish, or preparation of the manuscript.

### Grant Disclosures

The following grant information was disclosed by the authors:
Kaohsiung Chang Gung Memorial Hospital: CORPG8F1481, CMRPG8A0961, CMRPG8D1421-3, CMRPG8C0071, CMRPG8B0971-3, CMRPG8C0071, CMRP100-0689B, CMRPG8E1472.

### Competing Interests

The authors declare there are no competing interests.

### Author Contributions

- Ming-Hsien Tsai conceived and designed the experiments, analyzed the data, prepared figures and/or tables, authored or reviewed drafts of the paper, approved the final draft.
- Hui-Ching Chuang, Yu-Tsai Lin and Fu-Min Fang performed the experiments, authored or reviewed drafts of the paper, approved the final draft.
- Hui Lu contributed reagents/materials/analysis tools, prepared figures and/or tables, authored or reviewed drafts of the paper, approved the final draft.
- Wei-Chih Chen performed the experiments, prepared figures and/or tables, authored or reviewed drafts of the paper, approved the final draft.
- Chih-Yen Chien conceived and designed the experiments, performed the experiments, analyzed the data, authored or reviewed drafts of the paper, approved the final draft.

### Human Ethics

The following information was supplied relating to ethical approvals (i.e., approving body and any reference numbers):

This study was approved by the Medical Ethics and Human Clinical Trial Committees at Chang Gung Memorial Hospital (Ethical Application Reference number: 201701238B0).

### Supplemental Information

Supplemental information for this article can be found online at http://dx.doi.org/10.7717/peerj.4490#supplemental-information.

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
