# Peer review of "Clinical impact of albumin in advanced head and neck cancer patients with free flap reconstruction—a retrospective study"

_PeerJ, doi:10.7717/peerj.4490_

## Round 0.1 · original submission · Major Revisions

Dear Dr. Chien,

Please address the issues raised by the expert reviewers.

In addition, the following issues need to be addressed as well:

1) In lines 49,50 and 55, the cited reference should be placed before the period punctuation mark. Please correct this wherever necessary throughout the manuscript.

2) In line 202, please correct "clinician" as "clinicians".

3) If you prefer to keep Table 4 in the manuscript, please clearly define the unit of time of hospital stay (e.g. days, weeks, etc). Alternatively you may also define the unit in the manuscript in lines 139-141.

Best Regards,

Reviewer 1 ·

Basic reporting

In this study, the authors aim to determine the prognostic value of serum albumin level among 233 patients with advanced head and neck squamous cell carcinoma (HNSCC) were diagnosed with HNSCC and underwent surgical resection of tumor with microvascular free flap reconstruction simultaneously between January 2009 and December 2011 in Kaohsiung Chang Gung Memorial Hospital, Taiwan. I think that the manuscript might be interesting for the researchers who are mainly interested in head and neck cancer. In general, the manuscript is clearly written and the tables are well organized. However, the paper needs some additional works, in particular the statistical methods parts, and some minor changes are recommended before considering the paper for publication.

Experimental design

My concerns are as follows:

In abstract,

In Line 25-28: Please provide details for statistical methods in methods section of the abstract.

In materials and methods, statistical analysis

In Line 91-102: The authors also did not provide which normality test they used to check the normality assumption of the data. The following normality tests, such as Shapiro-Wilk test, Kolmogorov-Smirnov normality test or Anderson-Darling test can be thought to assess normality.

In Line 92: Please change “SPSS 20.0” to “IBM SPSS Version 20.0”.

In Line 95-96: Student’s t-test was used to compare the means of independent samples. Thus, the sentence “An independent sample t-test was used to analyze the length of hospital stay after surgery.” should be corrected. For example, authors can use “An independent sample t-test was used to compare the mean length of hospital stay after surgery for two patient groups of postoperative albumin levels (≥3.5 versus <3.5)”.

In Line 128: The authors mentioned that they used chi-square test, univariate and multiple regression analysis. However, they did not provide these tests in the “statistical analysis” part. Please update this part.

In Table 1,

Please change “Median Age [range] (year)” to “Median Age (year) [range]”.

Please change “Median follow up time [range] (year)” to “Median follow up time (year) [range]”.

Please recalculate the percentages of “recurrent site”, because the total number of these values does not equal to 100%. Use total number of recurrence (75 cases) to calculate these percentages rather than 233 cases. In addition to this, please correct the recurrent site of the patient (no:106) as “local” in raw data. Then, the total number of local recurrent site will be 26 as in Table 1.

In Table 4,

The table 4 may be removed, these findings can be interpreted in the results part.

Validity of the findings

no comment

·

Basic reporting

The study is about the determination of the prognostic value of the serum albumin level for the patients with advanced HNSCC (head and neck squamous cell carcinoma) undergoing surgery with simultaneous free flap reconstruction. In the analyses, the authors use the multiple regression to find the relation between clinical factors and postoperative major wound infection.

Regarding the analyses and associate conclusion, I consider that the article can be published once the listed suggestions will be made.

On the other side, I think that the problem is well-defined and literature is enough to cite the previous works.

Experimental design

The experimental set-up is well-defined and related works are listed. Whereas I consider that the analyses should be renewed partially. Because in the analyses, the authors directly use pairwise p-value between each clinical factor versus postoperative major wound infection (Table 2). But the interrelation between each clinical factor is not be considered. I think that the correlation between factors should be computed initially. From the results, if a high multicollinearity is observed, the analyses should be revised based on the independent variables (i.e., factors). Furthermore, the authors use the univariate analysis. But I think that all factors should be taken as predictors and the response is chosen as postoperative major wound infection under the multiple regression as long as the assumptions of this model are checked. Then the significance of this big model (each predictor in the model) can be evaluated for their significance. Here, if the assumptions of the multiple regression cannot be maintained, its non-parametric version can be applied too.

Moreover, I think that rather than the individual t-test, the Bonferroni corrected t-test can be used as the analyses are done under simultaneous effects of all risk factors.

Validity of the findings

Regarding the suggested analyses in Section 2, the results can be renewed.

Additional comments

None

---

## Round 0.2 · accepted · Accept

Dear Dr. Chih-Yen,

I am glad to inform you that your manuscript is now acceptable for publication.

However, the following typos should be fixed while in Production:

1) In line 29, "were used" should be used instead of "was used"

2) In line 33, "analyse" should be used instead of "analyses"

3) In line 140, please use "This analysis showed that" instead of "The demonstration shown us".

Best Regards,

Reviewer 1 ·

Basic reporting

no comment

Experimental design

no comment

Validity of the findings

no comment

Additional comments

no comment